# Wheat individual grain-size variance originates from crop development and from specific genetic determinism

**Aurore Beral** **\***, **Renaud Rincent, Jacques Le Gouis, Christine Girousse, Vincent Allard**

UMR 1095 GDEC, INRAE, Université Clermont Auvergne, Clermont-Ferrand, France

\* aurore.beral@inrae.fr

## Abstract

Wheat grain yield is usually decomposed in the yield components: number of spikes / m$^2$, number of grains / spike, number of grains / m$^2$ and thousand kernel weight (TKW). These are correlated one with another due to yield component compensation. Under optimal conditions, the number of grains per m$^2$ has been identified as the main determinant of yield. However, with increasing occurrences of post-flowering abiotic stress associated with climate change, TKW may become severely limiting and hence a target for breeding. TKW is usually studied at the plot scale as it represents the average mass of a grain. However, this view disregards the large intra-genotypic variance of individual grain mass and its effect on TKW. The aim of this study is to investigate the determinism of the variance of individual grain size. We measured yield components and individual grain size variances of two large genetic wheat panels grown in two environments. We also carried out a genome-wide association study using a dense SNPs array. We show that the variance of individual grain size partly originates from the pre-flowering components of grain yield; in particular it is driven by canopy structure via its negative correlation with the number of spikes per m$^2$. But the variance of final grain size also has a specific genetic basis. The genome-wide analysis revealed the existence of QTL with strong effects on the variance of individual grain size, independently from the other yield components. Finally, our results reveal some interesting drivers for manipulating individual grain size variance either through canopy structure or through specific chromosomal regions.

## Introduction

It is estimated that over the 28-year period 1980 to 2008 atmospheric warming has lowered wheat yield by about 5.5% globally [1]. This yield decline is expected to continue, with climate change predicted to further increase mean temperatures and also the frequencies and magnitudes of extreme weather events [2]. A number of authors have stressed the need for a better understanding of wheat crop yield physiology, to enable us to achieve the required improvements in wheat yield into the future [3–5].

**Data Availability Statement:** Genotypic files concerning the first genotypic panel (S1 Table) are available from the Data verse database (https://doi.

org/10.15454/MB4G3T). Phenotypic files are available from the Data verse database (https://doi.org/10.15454/5MR0EI). Genotypic files concerning the second genotypic panel (S2 Table) are available from the Science advances database (https://advances.sciencemag.org/content/suppl/2019/05/23/5.5.eaav0536.DC1, S2 Fig)

**Funding:** This study was supported by the Agence Nationale de la Recherche of the French government through the programme "Investissements d'Avenir" (16-IDEX-0001 CAP 20-25). A. Beral PhD Scholarship was funded by CAP 20-25 and INRA AgroEnv division.

**Competing interests:** The authors have declared that no competing interests exist.

Grain yield in bread wheat is usually analysed through various yield components: the number of spikes per $m^2$, the number of grains per spike, the number of grains per $m^2$ and the thousand-kernel weight (TKW). Strong negative relationships between these yield components have already been found for both environmental and genetic sources of variation [6], in particular between (i) the number of spikes per $m^2$ and the number of grains per spike and (ii) the number of grains per $m^2$ and TKW. The number of grains per $m^2$ (the product of the number of spikes per $m^2$ and the number of grains per spike) is determined before anthesis and has been shown to be the main determinant of grain yield [7], particularly under optimal conditions [8].

However, under more limiting conditions during the reproductive phase, TKW may also be strongly affected, since short periods of high temperature (>35°C) often occur during grain-filling, and these periods will likely increase with climate change [9]. Indeed, abiotic stress during grain-filling has been shown to affect TKW strongly [10–12]. Depending on the nature, duration and intensity of abiotic stress events, the grain-filling parameters are affected differently. For example, post-anthesis drought reduces the final grain mass by shortening both the duration and the rate of grain-filling [12]. For heat stress, moderate to high temperatures tend to shorten grain-filling duration and to accelerate the rate of grain-filling, but without full compensation [13,14]. Therefore, the TKW and its development from anthesis to maturity, becomes of high interest when studying post-anthesis abiotic stress, since any genetic variability associated with this trait could represent an interesting source of tolerance to extreme climatic events whose occurrence will likely increase in the future.

The TKW represents the bulk mean individual grain mass. It is usually studied at plot scale, but without consideration of possible changes in the underlying variability of individual grain mass. This intra-bulk variability results from the fact that grains of the same bulk sample (a plot of a single genotype, in a given environment) originate from a population of different spikes, of spikelets in each spike and of florets in each spikelet. This variability has long been observed in many species, both wild [15] and cultivated [16–21]. It is moreover present at every scale: inter-population, intra-population, intra-individual, and intra-organ (in our case, spike) [22,23].

Whatever its origins, micro-environmental or ontogenetic or stochastic, the variability in the mature masses of individual grains is assessed differently, depending on context. In an ecological context with wild species, the variability of individual mature grain mass is often considered an adaptative trait. Indeed, associated with variability of seed performance—e.g. seed vigour [22], seedling growth [24,25], tolerance to stress [26] - a large variability in mature seed mass allows species to adjust to unpredictable environmental conditions, so enhancing survival [23]. In contrast, in an agronomic context, and for bulk-harvested crops such as wheat, high variability of mature grain mass is often considered a disadvantage. Thus, when grains undergo processes such as milling, high variability in grain mass adversely affects the nutritional and technological quality of the end-products. Similarly, when grains are sown for a new crop, mass variability increases the heterogeneity of germination and of seedling emergence, often leading indirectly to decreased ease of crop management and decreased yield [27]. For this reason, breeders have usually sought for greater homogeneity in individual mature grain mass in seed lots [27,28].

In the agronomic context, and given the likely increase in abiotic stress events with climate change, it is worthwhile to better understand how individual mature grain mass variability comes about. In particular, because the TKW is developed gradually during crop development, the question arises as to how much of the individual mature grain mass variance is related to the various yield components and how much to specific genetic factors.

In recent years, there has been an increase in research into genetic determinism of trait variance between environments (plasticity) [29,30]. For example, Kusmec *et al.* [29] have shown in maize that genetic control of the plasticity or the means of a set of 23 traits (including plant height, ear length, days to silk, total kernel volume etc...) are independent of one another. This suggests it should be possible to generate new plant varieties "with both high mean phenotype values and levels of plasticity that are appropriate for their target performance environments". Conversely, Li *et al.* [30] have shown for kernel weight- and size-related traits, a moderate to high common genetic control exists for both plasticity (between environments) and variability (within an environment). To date, no similar studies have been conducted either on the variance of wheat TKW within an environment nor on the components of yield.

Hence, the objectives of this study are: (i) within two large genetic panels to describe the genetic variability, of the variance of individual mature grain size (using grain projected area as a proxy for individual grain size); (ii) To characterise the links between the variance of individual mature grain size and mean yield components; and (iii) To explore for the first time the genetic determinism of the variance of individual mature grain size.

## Materials and methods

### Field experiments

The field experiments were conducted at INRA, Clermont-Ferrand (45˚78' N, 3˚08' E, 401 m a.s.l) in 2015/2016 and 2016/2017 using the Pheno3C high-throughput, field-phenotyping platform.

Two different genetic panels of winter wheat were used in 2015/2016 and 2016/2017. Hereafter, these will be described, respectively, as 'panel 2016' and 'panel 2017'. Panel 2016 was composed of 228 elite genotypes (S1 Table) chosen to represent genetic variability within a pool of winter wheat adapted to western European conditions. This panel contained elite genotypes released between 1974 and 2014 [31–33]. Panel 2017 was a larger genetic panel of 312 genotypes (S2 Table) chosen to represent the worldwide genetic diversity of winter wheat, with some filtering on plant height and heading time to eliminate the more extreme genotypes. Panel 2016 was sown on 12/11/2015 and harvested between 11/07/2016 and 21/07/2016; and panel 2017 was sown on 4/11/2016 and harvested between 28/06/2017 and 17/07/2017. In all cases the sowing density of 250 grains m$^{-2}$ followed local agricultural practice.

The Pheno3C facility consists of eight plots of 96 micro-plots. Four plots can be covered by automatic rain-out shelters to generate the required levels of water stress without further modification of the micro-environment. The four remaining plots were control plots maintained under the local climate, but that could be irrigated to minimise water stress. Each micro-plot had a harvestable area of 2.1 m$^2$ and was managed following local agronomic practices. In the two experiments, two conditions were imposed: one irrigated control condition and one drought condition under rain-out shelters. For our current purposes, the water stress was not considered as such but merely as a source of environmental variability to ensure the results were not too environmentally specific and so would allow a degree of generalisation. The rainfall deficits in the water-stressed plots, compared to the control plots, were 220 and 240 mm in 2016 and 2017, respectively (S1 Fig). In both trials, the soil water in the control plots was nearly always easily accessible, whereas in the drought plots plants were exposed to some degree of water stress from approximatively mid-April onwards (S1 Fig). The two seasons x two water deficit conditions were considered to be independent environments and are referred to hereafter as E1 to E4 where: E1 (well-watered, 2016), E2 (water-deficit, 2016), E3 (well-watered, 2017) and E4 (water-deficit, 2017).

Given the large number of genotypes tested in these two genetic panels, they could not all be replicated. The experimental plan for panel 2016 was based on an augmented design as follows: each plot of the facility was considered as a block composed of two sub-blocks (Northern and Southern halves). So, each treatment is composed of eight sub-blocks grouped by pairs in four blocks. Each sub-block received genotypes of similar earliness. There were three replicates per block for four control genotypes. Among the non-control genotypes, 164 had no replicates, 32 had one replicate within the same sub-block, and 32 had one replicate in the adjacent sub-block. The experimental plan for panel 2017 was slightly different. For both conditions, the trial was divided for both conditions into eight sub-blocks of earliness (as for panel 2016), four control genotypes were sown in each sub-block, and there was no replicate for the other 308 genotypes. The genetic panels were very different, with no genotype being common between them. Three out of the four controls were common to both seasons.

## Phenotypic measurement and genotypic data

**Phenotypic data.** In each environment, the number of spikes per $m^2$ (SPM2) was determined after anthesis, by manual counting all spikes in two, 1-m long transects per micro-plot. At physiological maturity, all micro-plots were harvested using a combine harvester and the yield components (detailed below) were determined from the harvested grain samples, referred to hereafter as 'bulk grain'. Grain humidity was measured using a humidimeter (TM, Tripette and Renaud, France) to calculate grain mass yield at a notional 15% humidity basis.

For each grain bulk, a 200 g subsample was oven-dried (60°C for 48 h) before determining TKW, with an automatic seed-weighing and counting device (Opto Agri2, Optomachine, France). From direct measurements of yield (GY), SPM2 and TKW, the numbers of grains per spike (GPS), and numbers of grains per $m^2$ (GPM2) were computed.

The device used for TKW determination also provides single-grain-size components. Individual grain size (projected area) was extracted and used as a proxy for individual grain mass (S2 Fig and S3 Fig). Depending on the bulk grain and the season (2016 or 2017), the analyses of individual grain size distributions were based on 300 to 500 individual grains (S3 Table).

**Genotypic data.** The 228 genotypes from the 2016 trial and the 312 genotypes from the 2017 trial were genotyped with the TaBW280K SNP genotyping array [34]. Markers with missing values above 10%, minor allele frequency above 5% and monomorphic markers were removed. Imputation of missing values was carried out with Beagle V4.1 software [35]. The physical map used was the RefSeqV1.0 reference sequence of Chinese Spring [36].

## Statistical analyses

**Descriptors of the distributions of individual grain size.** To compare grain-size distributions between genotypes and environments, different metrics were calculated. For each bulk grain sample, an individual grain-size distribution was described by the following metrics: mean, variance, and the 5th and 95th percentiles (respectively, P5 and P95). To increase readability, the two main metrics considered in the following analyses are abbreviated as: grain size mean (GSM) and grain size variance (GSV).

**Spatial adjustment of phenotypic data.** Adjusted means were calculated to correct each trait for spatial heterogeneity and to take the trial design into consideration. For each trait (yield components, P5, P95, GSV and GSM) in each environment, adjusted means for panels 2016 and 2017 were estimated by taking into account the spatial heterogeneity and the different earliness blocks using the following linear mixed model:

(1) $Y_{ijkl} = \mu + SB_j + B_k + G_i + \epsilon_{ijkl}$, where $Y_{ijkl}$ is the observation for a given trait for genotype i, sub-block j, block k and a possible repetition l, $\mu$ is the intercept, $SB_j$ is the random effect of

sub-block j, $B_k$ is the fixed effect of block k, $G_i$ is the fixed effect of genotype i and $\epsilon_{ijkl}$ is the residual following a normal distribution $N(0, \sigma_\varepsilon^2)$.

To estimate the broad-sense heritability of each trait, in each environment, the following linear mixed models were used:

(2) For the 2016 genetic panel: $Y_{ijkl} = \mu + \underline{SB(check)}_j + B_k + G_i + \epsilon_{ijkl}$, where $Y_{ijkl}$ is the observation for a given trait for genotype i, sub-block j, block k and repetition l, $\mu$ is the intercept, $\underline{SB(check)}_j$ is the random effect of sub-block j estimated with the controls only, $B_k$ is the fixed effect of block k, $G_i$ is the random effect of genotype i and $\epsilon_{ijkl}$ is the residual following a normal distribution $N(0, \sigma_\varepsilon^2)$.

(3) For the 2017 genetic panel, we proceeded in two steps because we only had the control genotypes to correct for spatial effects: in a first step, the four control genotypes were used to estimate the spatial effects with the following model $Y_{ijk} = \mu + SB_j + B_k + G_i + \epsilon_{ijk}$, where $Y_{ijk}$ is the observation for a given trait for check i, block k, and sub-block j. The phenotypes adjusted for the spatial effects (Ycorr) were used in a second step to estimate heritability with the following model: $Ycorr_i = \mu + \underline{G_i} + \epsilon_i$ where $\mu$ is the intercept, $\underline{G_i}$ is the random effect of genotype i and $\epsilon_i$ is the residual following a normal distribution $N(0, \sigma_\varepsilon^2)$.

Pearson's correlations were used to analyse the links between final yield components and GSV. Comparisons of Pearson's correlations between environments were made to assess the generalisation of the relations between final yield components and GSV.

To assess the differences between environments in terms of genetic variability associated with the descriptors of individual grain size distributions (GSM, GSV, P5 and P95), coefficients of variation of these metrics were calculated as a ratio between the standard deviation and the mean of each metric, for each environment.

**Genome-wide association (GWA) analyses.** For each environment, 269,421 SNP (2016 genetic panel) and 178,277 SNP (2017 genetic panel) were tested on seven traits using the adjusted means of, respectively, 228 genotypes (2016 genetic panel) and 312 genotypes (2017 genetic panel).

For each genetic panel the critical linkage disequilibrium (LD) was assessed according to the method described in Breseghello and Sorrels [37]. In addition, to prune the numbers of SNP by removing the most redundant, the LD of each pair of markers was estimated and one SNP of the pair was removed if the LD was greater than 0.9.

GWA mapping was carried out using the package GenABEL_1.8–0 [38] with a mixed model:

(2) $Y = \mu + X\beta + G + E$ Where Y is the vector of adjusted phenotypic values, $\mu$ the intercept, $\beta$ the additive effect of the tested SNP, X the incidence matrix and G and E are the random polygenic and residual effects, respectively. For each chromosome, polygenic effects are assumed to be normally distributed $N(0, \sigma^2 K_{chr})$, where $K_{chr}$ represents the relationship matrix based on the LOCO (Leave-One-Chromosome-Out) model [39]. We retained only the relationship matrices with no structure corrections. Cormier *et al.* [40] used 72% of the 2016 elite panel and did not find any structure. The 2017 genetic panel was optimised to be the less-structured. The results were concatenated after running the GWAS model for each trait and for each environment.

Next, we retained only the significantly associated SNP with LOD scores above three. The objective was to identify QTL of GSV associated with either both GSV and mean yield components, or with GSV only. To define QTL boundaries from the GWAS results, we used the method described in Cormier et al. [40]. Briefly, this is as follows. The first step is to develop a qualitative definition of the boundaries of presumed QTL, using visual assessment on Manhattan plots. The second step was to define the initial QTL boundaries. The LD was computed between every significantly associated SNP on all traits of interest (GSV and mean yield components) for each environment. A group of significantly associated SNPs belonging to the

same LD cluster was defined as the LD block. Clusters of LDs were defined by an average distance with a cut-off (1-"critical LD"). The initial QTL boundaries were defined with the LD blocks found in the visual presumed QTL. The third step was to extend the initial boundaries using the LD decay [40]. QTL with overlapping LD blocks were considered to belong to the same locus. Colocalisations between QTL of the different environments were assessed when these QTL shared at least one significantly associated SNP.

Linear regression was used to compare the percentages of total variance of GSV explained by QTL of GSV, with or without yield components. To avoid redundancy, for each QTL only the significant SNPs with the strongest effects were retained. Then, for each environment, linear regressions between GSV and QTL effects were carried out and the variances explained by each category of QTL (with or without yield components) were calculated. Finally, for each environment, the QTL explaining most of the total variance of GSV were identified using a stepwise regression model ($r^2_{opt}$) AIC based to determine the number of QTL which explained the most of GSV.

## Results

### Individual grain size variance is subjected to both genetic and environmental variabilities

Across all genotypes and environments, individual grain size (projected area) ranged from 2.67 to 29.98 mm$^2$. This illustrates the wide range of variation of grain size. Small differences in size were observed between environments for minimum grain size, ranging from 2.67 mm$^2$ (E4) to 5.54 mm$^2$ (E3) while the maximum grain sizes were very similar (S3 Table).

Overall, the variance of individual grain size varied more than the mean over a genetic panel, as GSV showed higher genetic variability than GSM (Table 1). As the variance of GSV could be higher than the variance of GSM by construction, comparison of coefficients of variation of the standard deviation (square root of GSV) and GSM were also carried out and these confirmed the previous result. Differences between environments were also observed (Table 1). Coefficients of variation of the GSM and the GSV were higher for E3 and E4, than for E1 and E2. This was probably caused by the higher genetic diversity of panel 2017 (E3 and E4).

### Individual grain size variance is partially driven by yield components

The four environments can be characterised by their mean yield components (Table 2). The number of grains per m$^2$ and, consequently, the canopy structure were different for the four environments. E3 (2017) had the highest number of grains per m$^2$ while E4 (2017) had the

**Table 1. Description of the distributions of individual grain size (projected area (mm$^2$)).**

|  | GSM | | | GSV | | | P95 | | | P5 | | |
|---|---|---|---|---|---|---|---|---|---|---|---|---|
|  | Mean | SD | CV | Mean | SD | CV | Mean | SD | CV | Mean | SD | CV |
| E1 | 17.10[c] | 0.93 | 5.44% | 7.30[c] | 1.48 | 20.27% | 20.94[c] | 1.17 | 5.59% | 12.23[b] | 0.90 | 7.36% |
| E2 | 16.95[c] | 0.98 | 5.78% | 6.21[b] | 1.25 | 20.13% | 20.50[b] | 1.19 | 5.80% | 12.48[b] | 0.87 | 6.97% |
| E3 | 16.56[b] | 1.35 | 8.15% | 7.06[c] | 2.08 | 29.46% | 20.23[b] | 1.67 | 8.26% | 11.70[a] | 1.22 | 10.43% |
| E4 | 15.43[a] | 1.37 | 8.88% | 4.90[a] | 1.16 | 23.67% | 19.11[a] | 1.50 | 7.85% | 11.72[a] | 1.15 | 9.81% |

For each environment, the mean, standard deviation (SD) and coefficient of variation (CV) of grain size mean (GSM), grain size variance (GSV), 95[th] percentile (P95) and 5[th] percentile (P5) were calculated and compared between environments. Individual grain projected area (mm$^2$) was used as a proxy for individual grain size (mass). E1 (well-watered, 2016); E2 (water-deficit, 2016); E3 (well-watered, 2017); E4 (water-deficit, 2017).

[a,b,c] values with the same letter within a column were not significantly different (P>0.05) according to a Tukey post-hoc test following ANOVA.

**Table 2. Description of yield components.**

|  | SPM2 (Number of spikes per m²) | | GPS (Number of grains per spike) | | GPM2 (Number of grains per m²) | | TKW (g 15% hum.) | | GY (t/ha 15% hum.) | |
|---|---|---|---|---|---|---|---|---|---|---|
|  | Mean | SD | Mean | SD | Mean | SD | Mean | SD | Mean | SD |
| E1 | 609[d] | 71 | 32[a] | 5 | 19062[c] | 2605 | 45.9[c] | 3.68 | 8.7[c] | 10.39 |
| E2 | 401[a] | 52 | 41[c] | 6 | 16267[b] | 2237 | 47.2[d] | 3.95 | 7.6[b] | 9.00 |
| E3 | 555[c] | 89 | 39[b] | 9 | 21638[d] | 4786 | 41.1[b] | 4.84 | 8.8[c] | 15.45 |
| E4 | 445[b] | 84 | 33[a] | 8 | 14187[a] | 2304 | 38.9[a] | 4.65 | 5.4[a] | 7.61 |

For each environment, the mean and standard deviation (SD) of all yield components were calculated and compared between environments.

SPM2: number of spikes per m², GPS: number of grains per spike, GPM2: number of grains/m², TKW: thousand kernel weight (g at 15% moisture content), GY: Grain yield (t/ha at 15% moisture content).

E1 (well-watered, 2016); E2 (water-deficit, 2016); E3 (well-watered, 2017); E4 (water-deficit, 2017): $R^2 = 0.89$.

Values with the same lowercase letter within a column are not significantly different (P = >0.05) based on a Tukey post-hoc test following ANOVA.

lowest. The environments showed different combination of spikes per m² and number of grains per spike to set the canopy structure. E1 (2016) had the highest number of spikes per m² and the lowest number of grains per spike. Conversely, E2 (2016) had the lowest number of spikes per m² and the highest number of grains per spike. Thus, each of the environments was represented by a uniquely different canopy structure with potentially different effects on GSV.

To analyse whether GSV was linked to average yield components, Pearson's correlations were calculated between GSV and each yield component (SPM2, GPS, GPM2, TKW, GY) (Fig 1). In each environment, GSV was positively correlated with TKW (which is equivalent to GSM) (S3 Fig). In addition, GSV was negatively correlated with SPM2. Some correlations appeared to be environment-specific. In E1, GSV was also positively correlated with GPS. In E2, E3 and E4, GSV was negatively correlated with GPM2. Since SPM2, GPS and GPM2 are components that are set up before flowering, *i.e.* before the determinism of grain size, the correlation of GSV with those traits indicates GSV is partly driven by pre-flowering processes and not only by differences in grain-filling processes. In addition, the link between GSV and the pre-flowering traits is independent of the environment and probably has a genetic nature. Pearson's correlations between SPM2 and GSV, as well as between TKW and GSV, were not significantly different among the environments (the correlations between SPM2 and GSV, p-values were 0.32 and 0.58 for the 2016 panel and 2017 panel respectively; the correlations between TKW and GSV, p-values were to 0.94 and 0.35 for the 2016 panel and 2017 panel respectively).

Within each genetic panel, Pearson's correlations between the GSVs of the two environments were high: 0.68 and 0.62 for, respectively, E1 *vs* E2 and E3 *vs* E4. These results emphasise that genotype-environment interactions are low and suggest a specific genetic determinism of GSV.

## Individual grain size variance also has a specific genetic determinism

Genetic determinism of GSV was studied through a genome-wide association study carried out on this particular trait and yield components. The focus was placed on the differentiation between GSV QTL that colocalised with yield components ("driven" QTL) from those which did not colocalise (specific QTL).

Globally, 4,895 SNP were significantly associated with GSV and 9,517 SNP with the other traits studied (yield components) including 1,555 SNP with SPM2, 1,614 SNP with GPS, 2,954 SNP with GPM2 and 3,394 SNP with TKW.

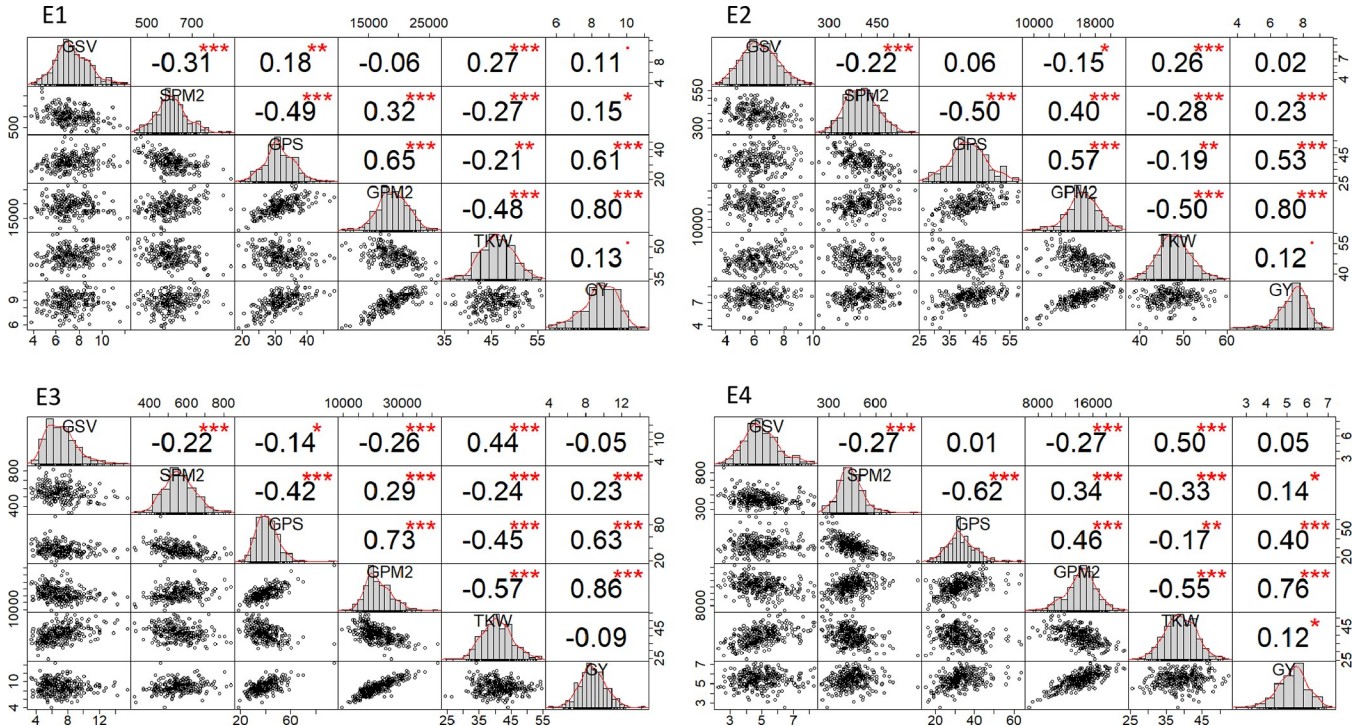

**Fig 1. Pearson correlations among grain size variance and yield components.** The diagonal panels show histograms for each trait. The lower and upper triangular panels, respectively, show scatter plots and Pearson correlation coefficients between the two traits. GSV: Grain size variance, SPM2: number of spikes per m², GPS: number of grains per spike, GPM2: number of grains/m², TKW: thousand kernel weight (g at 15% moisture content), GY: Grain yield (t/ha at 15% moisture content). E1 (well-watered, 2016); E2 (water-deficit, 2016); E3 (well-watered, 2017); E4 (water-deficit, 2017). ‘.’: P-value<0.1; ‘*’: P-value<0.05; ‘**’: P-value<0.01; ‘***’: P-value<0.001.

Sixty-four QTL were associated with GSV with a mean size of 95 Mb on the wheat reference sequence (Table 3). The whole results with the number of associated SNP for each chromosomal zone and the effects of the minor allele are available as supplementary data (S4 Table). The number of QTL by wheat sub-genome, was consistent with the SNP density, as 33 QTL were found on sub-genome A, 20 QTL on sub-genome B and only 11 QTL on sub-genome D (Table 3).

**Table 3. Description of the QTL found in the four environments.**

| Environment | QTL | Size (Mb) | | MAF[1] | | Number of QTL[2] | | |
|---|---|---|---|---|---|---|---|---|
| | Total | Mean | SD | Mean | SD | A | B | D |
| E1 | 16 | 161 | 245 | 0.24 | 0.14 | 10 | 4 | 2 |
| E2 | 21 | 48 | 153 | 0.27 | 0.14 | 11 | 8 | 2 |
| E3 | 13 | 66 | 144 | 0.28 | 0.10 | 6 | 4 | 3 |
| E4 | 14 | 104 | 156 | 0.25 | 0.13 | 6 | 4 | 4 |
| Total number of QTL | 64 | | | | | 33 | 20 | 11 |

For each environment, QTL associated with GSV were identified and their characteristics were reported.

E1 (well-watered, 2016); E2 (water-deficit, 2016); E3 (well-watered, 2017); E4 (water-deficit, 2017).

[1] Minor allele frequency (MAF)

[2] Number of QTL for each sub-genomes

**Table 4. Colocalisations of QTL.**

| | | Intra-population | | | | Inter-population |
|---|---|---|---|---|---|---|
| | QTL | QTL in one environment | | QTL common to two environments | | |
| | | 2016 Panel (E1 or E2) | 2017 Panel (E3 or E4) | 2016 Panel (E1 and E2) | 2017 Panel (E3 and E4) | |
| Specific QTL | 34 (53.1%) | 15 | 13 | 2 | 2 | 2 |
| Driven QTL | 30 (46.9%) | 12 | 9 | 6 | 2 | 1 |
| Total QTL | 64 (100.0%) | 49 (76.6%)[(1)] | | 12 (18.8%)[(1)] | | 3 (4.6%)[(1)] |

For each of the 64 QTL, colocalisations with SNP associated with GSV in other environments (common QTL) were identified. Then the presence ("Driven" QTL) or the absence (Specific QTL) of colocalisations with SNP associated with yield components were determined.

[(1)] Percentage of the total number of QTL

For each GSV QTL, colocalisations were checked with (i) SNP associated with yield components in the same environment and (ii) SNP associated with GSV in other environments, to find common QTL between environments (Table 4). Among all the 64 QTL for GSV, 34 (53.1%) QTL did not colocalise with any yield components (Table 4) and were considered as specific QTL. Among the 30 QTL for GSV which colocalised with yield components ("driven" QTL), GSV colocalised with either one or several yield components. Indeed 3 QTL colocalised with all yield components and 19 colocalised with only one yield component (S4 Table). The two populations pooled, there were 12 QTL common to two environments and three QTL common to the two different populations (Table 4), which underlines some robustness in the study. Only one QTL was common to both genetic panels.

Overall, for the four environments, GSV showed broad-sense heritabilities higher than SPM2 and GPS but lower than TKW (S5 Table). However, the heritability of GSV across the four environments was quite high, ranging from 0.52 to 0.85. This revealed that genetic factors play an important role for GSV. Furthermore, for each environment, all QTL explained between 40.1% and 49.0% of the total variance of GSV (Table 5). For the four environments, a total of 46 QTL were retained in the stepwise regression model and explained between 39.3% and 48.7% of the total variance of GSV (Table 5). Among these QTL, half of them colocalised with mean yield components ("driven" QTL) (Table 5). For all environments, specific QTL and "driven" QTL explained, respectively, between 18.3% and 23.5% and between 21.8% and 25.4% of the total variance of GSV explained by all QTL (Table 5). There was no difference between the two categories of QTL so, globally, the "driven" QTL did not explain more of the total variance of GSV, than specific QTL.

**Table 5. Explained variance by QTL.**

| Environment | All QTL | | Specific QTL | | "Driven" QTL | | Optimal number of QTL | | |
|---|---|---|---|---|---|---|---|---|---|
| | Number of QTL | $r^2$ | Number of QTL | $r^2$ | Number of QTL | $r^2$ | Number of QTL | With "Driven" QTL | $r^2_{opt}$ |
| E1 | 16 | 40.1% | 6 | 18.3% | 10 | 21.8% | 11 | 7 | 39.3% |
| E2 | 21 | 44.5% | 12 | 21.7% | 9 | 22.7% | 11 | 6 | 42.6% |
| E3 | 13 | 49% | 7 | 23.5% | 6 | 25.4% | 11 | 4 | 48.7% |
| E4 | 14 | 44.4% | 9 | 20.2% | 5 | 24.2% | 13 | 5 | 44.3% |
| Total | 64 | | 34 | | 30 | | 46 | 22 | |

For each environment, the number of QTL which colocalised with yield components ("Driven" QTL) or not (Specific QTL) were calculated. The percentage of the total variance of GSV explained ($r^2$) by both categories of QTL and by all QTL were then estimated using linear regression. For each environment, the QTL explaining most of the total variance of GSV (Optimal number of QTL) were identified using stepwise regression and the percentage of total variance of GSV explained ($r^2_{opt}$) by these QTL is reported.

E1 (well-watered, 2016); E2 (water-deficit, 2016); E3 (well-watered, 2017); E4 (water-deficit, 2017).

## Discussion

### Representativity of the environments studied

The main goal of the study was to examine the origins of the genetic determinism of individual grain size variance (GSV), using individual grain size (projected area, mm$^2$) as a proxy for individual grain mass. In particular, we wished to assess whether genetic variation for GSV is determined purely by canopy structure (i.e. correlated with yield components) or if it also displays a specific component (*i.e.* independent of canopy structure). For a certain degree of generalisation of our results to be attained, it is important that the environments studied allow us to highlight well-known relationships between traits, while also offering sufficient variation in the mean trait values.

As already exhaustively reviewed [6,41–44], we observed robust negative relationships between the main yield components in all four environments. The grain set trade-off was apparent in the negative relationship between the number of spikes per m$^2$ (SPM2) and the number of grains per spike (GPS). Similarly, the negative trade-off between the number of grains per m$^2$ (GPM2) and TKW was also clearly apparent in our dataset. The expected behaviour observed in our data was nevertheless accompanied by a large variability in mean trait values between environments. In particular, grain yield ranged between 5.4 and 8.8 t ha$^{-1}$ in E4 and E3, respectively. This range adequately samples the inter-annual yield variations observed in the Clermont-Ferrand area, when managed under local agronomic practices. These yield variations were driven mainly by large variations in canopy structure as shown by the 50% difference observed for SPM2 between the environments with the lowest (E1) and highest (E4) spike densities, respectively (Table 2). At the same time, the four environments displayed large variations in trait values and classical relationships between traits and therefore offer the desired genericity to support the main analysis of this work.

### GSV is partly driven by canopy structure

To understand to what extent GSV was driven by yield components, correlations between each trait were studied within each environment (Fig 1). In the beginning, our expectation was that any increase in the GPM2 (at the same plant density), through either an increase in GPS or in SPM2, would lead to an increase in GSV. Indeed, an increase in GPS created a larger proportion of small grains [45], mainly through an increase of the number of grains in the distal positions of each spikelet [46]. These distal grains have been shown to be significantly smaller than the grains in proximal positions [47,46,48,49]. In addition, an increase in SPM2 implies a higher diversity in the spike population at the plot scale, with a larger proportion of spikes from secondary tillers [50]. These have been shown to carry a higher proportion of small grains than the main tillers [45]. Clearly, our initial hypothesis of a positive correlation between GSV and any component of grain number (SPM2, GPS and GPM2) was invalidated by the results obtained (Fig 1). Indeed, the observed correlations between these three variables and GSV were clearly not positive (except for the correlation between GSV and GPS in E1).

Conversely, in all environments, our results demonstrate a negative correlation between both GSV and GPM2 and between GSV and SPM2. Indeed, an increase in SPM2 leads to an expected increase in GPM2 and to a decrease in TKW. These effects are accompanied by a negative effect on GSV. An increase in GPS generates an increase in GPM2 and a decrease in TKW but has no effect on GSV. However, both SPM2 and GPS show similar correlations with TKW, both in kind and in magnitude. The question then arises as to what process can explain the differential behaviour of GPS and SPM2 on GSV.

First, we examine the nature of the relationship between SPM2 and GSV. An increase in SPM2 profoundly changes canopy structure by increasing the proportion of small spikes. This is the result of an increase in secondary tillers in the spike population. As earlier mentioned, the secondary tillers contain a higher proportion of small grains [45] and these will likely lead to an increased GSV. However, our results conflict with this, so the increase in SPM2 must be accompanied by some other modification of the grain population, leading to a diminution of GSV. It seems likely that increasing SPM2 also led to a reduction in the size of the largest grains. This interpretation finds support in the negative correlation between the 95th percentile of individual grain size and SPM2 (S4 Fig). Conversely, an increase in GPS, while also increasing the proportion of small grains (S4 Fig), did not bring about a similar reduction in the size of the largest grains (except for E3, S4 Fig). This may explain the absence of correlation between GPS and GSV. This explanation would seem to find support in previous studies showing that fruiting efficiency of tiller spikes (defined as the number of grains per unit spike dry weight at anthesis [51]) is higher than in those of the main stem [52]. For a given level of source, the higher number of grains in the small spikes, should result in a decrease in the growth potential at anthesis of the bigger grains (as defined by Bremner and Rawson [53]) and thus their final size.

The above proposition shows the need to clarify the nature of the interactions caused by spike diversity within a canopy (GPS, spike morphology) and the variability of intra-spike individual grain size. However, the core of our study is to determine to what extent GSV is driven by canopy structure. Our results clearly show that a large part of GSV can be explained by SPM2. Nevertheless, the unexplained part of GSV could have to do with a specific genetic basis. This idea should be explored further.

## GSV also has a specific genetic determinism

Our results identify a specific genetic determinism for GSV that is highly heritable. In general, GSV heritability ranged between 0.52 and 0.85 (S5 Table). Heritability in GSV was higher than in SPM2 or GPS (except for E4) and lower than in TKW, in which heritability was the highest for any of the yield components (S5 Table). This hierarchy of yield component heritabilities is in line with the literature [54]. Moreover, our genome-wide analysis reveals that about one half of the QTL associated with GSV were specific (i.e. exhibiting no colocalisation with the SNPs of the yield components) (Table 4). These specific QTL contributed about half of the total variance of GSV, explained by all QTL for each environment—the range being between 40.1% and 49.0% of the total GSV variance (Table 5). QTL explained the same proportion of the total variance for GSV (specific and "driven") as for TKW. For example, Cormier *et al.* [40] showed that, on average, QTL explained 32.3% of the total variance of TKW. This emphasises that GSV has a genetic determinism as strong as the one governing TKW. As expected, GSV is a trait submitted to strong G*E interactions (Table 4). Approximately 18.8% of the QTL were found in common in the two environments where the two populations were tested, which is within the range of the environmental robustness classically observed for wheat quantitative traits [55,56]. Furthermore, three QTL were found in common between the two populations. The strong specificity of the QTL depending of the genetic background is again a classical observation [55].

The sensitivity of the QTL identified to both genetic background and environmental conditions are however not different between the two types of QTL ("driven" and specific).

Therefore, even though prone to genotype x environment interactions, our results strongly suggest that the genetic determinism of GSV has a specific nature in addition to common genetic bases with other yield components.

An extensive literature has investigated the genetic control of mean grain size, shape and weight [57–61]. A few have studied the genetic determinism of the dynamic parameters of mean grain-filling in wheat [62,63]. On the other hand, genetic studies are few at the single-grain level. To our knowledge, only one has explored genetic control at single-grain level in wheat by focusing on grains at specific positions within the spikelet (basal vs. distal) in a recombinant inbred line mapping population [63]. Nevertheless, this does not fully account for the variability of individual grain size at intra-genotype level. Our study is the first to deal explicitly with the genetic determinism of grain size variance.

On a much wider scale, the genetic determinism of biological trait variance (defined as the variability observed for a trait within an environment and not across different environments) has already received some attention. For example, there are numerous studies in animal breeding [64,65] that report either many QTL with small effects on trait mean independent of trait variance [64] or a few common QTL with strong effects for trait mean and variance [65]. The genetic determinism of traits or of gene expression have been explored in plants such as *Arabidopsis thaliana* [66,67]. These show trait variance is heritable and also that some QTL influence trait variability strongly and specifically, independently of mean value. The purposes of these studies have mostly been to identify and evaluate the genetic control of biological trait variance in order to select for specific trait variance.

A similar objective for individual grain size variance in wheat may be achieved through two levers.

1) We have already shown that GSV derives partly from genotype x environment interactions affecting canopy structure before anthesis and also, partly, from a genetic effect for a given structure at anthesis. Therefore, GSV may be manipulated by either modifying the canopy structure or by selecting for specific GSV without affecting the canopy structure. In the first case, canopy structure can be manipulated through a) the selection of genotypes that display specific yield-establishment strategies, and/or b) agronomic practices (sowing density, N fertilisation etc. . .) that affect yield components in such a way that a desired value of GSV is driven by canopy structure. Independent of strategy, genotype selection and practices to obtain relevant canopy structures appear feasible, since yield components are often measured in agronomic trials, and these processes can be speeded up using reliable, high-throughput phenotyping methods, in particular for SPM2, the most critical component [68].

2) The second lever is through direct manipulation of the specific genetic determinism of GSV. This is hindered by the supplementary workload involved in phenotyping individual grain variance. This hinderance applies even when automated image analysis of grain size is utilised. Nevertheless, being able to select for GSV without affecting canopy structure could be useful in some situations. For example, in Australia where severe post-flowering water stress occurs, tolerant genotypes have been selected by optimising canopy structure through the introduction of a tiller inhibition (tin) gene [69]. In this case, GSV manipulation could be achieved only by utilising the existing, specific genetic determinism of the trait. By construction, specific QTL of GSV have no direct impact on yield since they are not linked with any yield components, but may nevertheless be of interest for breeders to select for traits that are independent of yield.

## Breeding for GSV: Putative interest for wheat?

As mentioned above for animal models, manipulation of variance is of considerable interest in breeding, in particular to increase the homogeneity of a selected trait. In the specific case of bread wheat, grain size variation has economic implications. First, grain size has been shown to be positively correlated with specific grain weight [70], a trait related to the efficiency of

wheat transport and storage. And, second, a high proportion of small grains is penalised commercially by the milling industry because many small grains are lost during the cleaning process prior to milling. Also, a high proportion of small grains is indicative of a poor flour yield [71,72].

Increased grain homogeneity can also be justified on grounds of nutritional quality. For example, dry gluten content is variable between grains in bulk, mainly because of variation linked to their location in the spike—i.e. basal, median or apical [73]. Micro- and macro-nutrient content also varies between grains [74] –i.e. concentrations generally decrease from basal to apical spikelets. Thus, breeding for wheat with fixed (high) values of technological and nutritional quality, implies selection of genotypes having low proportions of small grains (high homogeneity).

When used as seed for a new crop, grain size homogeneity is also a positive trait. For wild species, some studies have emphasised the advantages of grain-size variation on germination, dormancy and dissemination success [15,23,75]. Thus, grain-size variability can represent a bet-hedging strategy which maximises the chances of generational survival in a fluctuating environment. However, for a cultivated species in an agricultural context, grain-size variability can lead to non-uniform crop establishment. Clearly, this will have negative outcomes in conventional agriculture.

Nevertheless, in the context of climate change and increasingly unpredictable weather, our negative perception of individual grain size variability may change to a more positive one. This is due to the putative positive and more reliable agronomical outcomes for this trait under some conditions [76]. Here, the link between grain size variability and crop establishment is worth re-examination. A greater variability in individual grain size, and thus in seed performance, may favour irregular seedling emergence, which may help buffer the impacts of environmental stress events during establishment [77]. This strategy may well come to sit alongside the more classical one of breeding for intrinsic stress tolerance.

We believe our results call for a more thorough examination of the correlations between individual grain size variability and mean traits of agronomic interest. In particular, we should explore the effects of different GSV on yield under post-flowering abiotic stress, since single grains have been shown to exhibit different responses to post-anthesis temperature stress, depending on their position on the spike or within a spikelet [78]. This should widen perspectives for breeding by using individual grain-size variability as a trait for increased tolerance of abiotic stress.

## Supporting information

**S1 Table. Genetic panel 2016.** Wheat varieties tested, year of release (YR) and geographical origin (Country).
(PDF)

**S2 Table. Genetic panel 2017.** Wheat varieties tested, year of release (YR) and geographical origin (Country).
(PDF)

**S3 Table. Descriptive analysis of bulk grain samples.** For each environment, descriptive statistics were calculated for all grains composing the bulk samples harvested on each micro-plot. E1 (well-watered, 2016), E2 (water-deficit, 2016), E3 (well-watered, 2017) and E4 (water-deficit, 2017).
(TIF)

**S4 Table. Description for each chromosomal zone of interest.** We identify and report the characteristics of each chromosomal zone of interest (64 QTL associated with grain size variance (GSV)). [1]Number of SNP associated with a trait in this chromosomal zone, [2]Maximum LOD score on the significant associated SNP, [3] Physical position on the chromosome (Chr) [36] [4]Minor allele frequency (MAF), [4]Expressed in terms of the absolute value for each trait. E1 (well-watered, 2016), E2 (water-deficit, 2016), E3 (well-watered, 2017) and E4 (water-deficit, 2017).
(PDF)

**S5 Table. Heritabilities of GSV and yield components.** For each environment, the broad sense heritabilities (see Materials and methods) of all yield components were calculated and compared between environments. GSV: Grain size variance, SPM2: number of spikes per m$^2$, GPS: number of grains per spike, GPM2: number of grains/m$^2$, TKW: thousand kernel weight (g at 15% moisture content), GY: grain yield (t/ha at 15% moisture content). E1 (well-watered, 2016), E2 (water-deficit, 2016), E3 (well-watered, 2017) and E4 (water-deficit, 2017).
(PDF)

**S6 Table. Description for each significant SNP related traits.** Characteristics (Environment (Env), Trait, Chromosome (Chr), Name of the SNP (SNP), Physical position, MAF, Effect, LOD score and Percentage of the total variance of each trait explained by the significant SNP) of each significant SNP are identified and reported. [1] Physical position on the chromosome (Chr) [36], [2]Minor allele frequency (MAF), [3]Expressed in terms of the absolute value for each trait, [4] LOD score on the significant associated SNP, [5] Percentage of the total variance of each trait explained (r$^2$). E1 (well-watered, 2016), E2 (water-deficit, 2016), E3 (well-watered, 2017) and E4 (water-deficit, 2017).
(PDF)

**S1 Fig. Cumulative rainfall (mm) and changes in the available soil water for the control (blue line) and stressed (red line) plots, for the two seasons (2016 and 2017).** Soil field capacity, limit of easily accessible water and permanent wilting point were calculated from soil water retention characteristics determined over the soil profile.
(TIF)

**S2 Fig. Individual grain size (projected area) in relation to individual grain mass for 24 micro-plots (n = 2,303).** The red diagonal represents the linear regression between individual grain projected area and individual grain mass (R$^2$ = 0.83).
(TIF)

**S3 Fig. GSM (grain size mean) in relation to TKW (thousand kernel weight) for the four environments.** The red diagonal represents the linear regressions between GSM and TKW for each of four environments. E1 (well-watered, 2016): R$^2$ = 0.81; E2 (water-deficit, 2016): R$^2$ = 0.83; E3 (well-watered, 2017): R$^2$ = 0.78; E4 (water-deficit, 2017): R$^2$ = 0.89.
(TIF)

**S4 Fig. Pearson correlations among grain sizes variance and yield components.** The diagonal panel shows histograms of each trait. The lower and upper triangular panels show, respectively, the scatter plot and the Pearson correlation coefficient between the two traits. P95: 95[th] percentile of grain sizes, SPM2: number of spikes per m$^2$, GPS: number of grains per spike. E1 (well-watered, 2016), E2 (water-deficit, 2016), E3 (well-watered, 2017) and E4 (water-deficit, 2017). '.': P-value<0.1; '*': P-value<0.05; '**': P-value<0.01; '***': P-value<0.001.
(TIF)

## Acknowledgments

The authors also acknowledge the projects BreedWheat (ANR-10-BTBR-03) and PHENOME (ANR-11-INBS-0012) for providing raw data. The authors want to thank, D. Cormier and B. Adam for managing the Pheno3C phenotyping platform, W. Ngo, L. Bonjean and members of UE PHACC for their technical assistance. The authors want also to thank R. Rincent and A. Mini for providing script and useful help regarding to adjusted means calculation and GWAS analysis. The authors thank Dr. Sandy Land (www. rescript.co.nz) for language editing and proofreading of the manuscript.

## Author Contributions

**Formal analysis:** Aurore Beral, Renaud Rincent.

**Funding acquisition:** Jacques Le Gouis, Christine Girousse, Vincent Allard.

**Supervision:** Jacques Le Gouis, Christine Girousse, Vincent Allard.

**Writing – original draft:** Aurore Beral.

**Writing – review & editing:** Renaud Rincent, Jacques Le Gouis, Christine Girousse, Vincent Allard.

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
