## [Decision Letter · Decision Letter 0]

9 Jan 2020

PONE-D-19-34899

Wheat individual grain-size variance originates from crop development and from specific genetic determinism

PLOS ONE

Dear Ms. Beral,

Thank you for submitting your manuscript to PLOS ONE. After careful consideration, we feel that it has merit but does not fully meet PLOS ONE’s publication criteria as it currently stands. Therefore, we invite you to submit a revised version of the manuscript that addresses the points raised during the review process.

We would appreciate receiving your revised manuscript by Feb 22 2020 11:59PM. To enhance the reproducibility of your results, we recommend that if applicable you deposit your laboratory protocols in protocols.io, where a protocol can be assigned its own identifier (DOI) such that it can be cited independently in the future. For instructions see: http://journals.plos.org/plosone/s/submission-guidelines#loc-laboratory-protocols

We look forward to receiving your revised manuscript.

Kind regards,

Aimin Zhang, Ph.D.

Academic Editor

PLOS ONE

Journal Requirements:

Reviewers' comments:

Reviewer's Responses to Questions

**Comments to the Author**

1. Is the manuscript technically sound, and do the data support the conclusions?

Reviewer #1: Yes

Reviewer #2: Partly

2. Has the statistical analysis been performed appropriately and rigorously? 

Reviewer #1: Yes

Reviewer #2: I Don't Know

3. Have the authors made all data underlying the findings in their manuscript fully available?

Reviewer #1: Yes

Reviewer #2: Yes

4. Is the manuscript presented in an intelligible fashion and written in standard English?

Reviewer #1: Yes

Reviewer #2: Yes

5. Review Comments to the Author

Reviewer #1: In this manuscript, authors used two large genetic wheat panels under two environments and investigated the individual grain size variances and yield components. They showed that the variance of individual grain size originates from the pre-flowering components of grain yield and also has a specific genetic basis,. Moreover, they identified 64 QTLs with the effects on the variance of individual grain size, either dependently or dependently from the other yield components, suggesting that individual grain size variance could be used as a trait to be manipulated in wheat breeding practices.

Overall, this manuscript provides another layer of insight into the intra-genotypic variance of individual grain mass and its effect on TKW. The data is sufficient to support their conclusions. However, a few of minor points should be considered to improve the manuscript.

Specific comments:

1. Authors identified a total 64 OTLs for variance of individual grain size and listed their chromosome distribution but without their position. I think it is better to give a rough position in each chromosome either physically or genetically.

2. Authors also mentioned they identified QTLs for yield components but did not show these QTLs. I think this information should be provided. Since extensive such QTLs have been reported, how many such QTLs are likely overlapping with reported?

3. Regarding to GSV, the 30 “driven” QTLs obviously contribute to the yield, how about 34 “specific” QTLs? The biological function of these specific QTLs should be discussed.

4. The manuscript should be more concise, especially Abstract, Introduction and Discussion parts.

Reviewer #2: Although this research seemed to be interesting, the manuscript should be further revised, especially the experiment desgin and data analysis, language, etc.

Introduction

p11: Line 108-113 and Line 114-122 seemed to be repetitive, so delete one of them or revise.

Experiment design:

The two different populations were used in two years under well-watered and water-deficit, in fact, each one population has been grown in one year under two treatments, so E1,E2 E3 and E4 should not be used to anlyze data. therefore recommend the authors to analyze the data under different treatment in one population, respcetively.

Why there are no replicates? It is not clear.

Results

Many SNPs were found in different traits, but the significant SNP related trait was not pointed out.

The language should be further revised.

6. PLOS authors have the option to publish the peer review history of their article (what does this mean?). If published, this will include your full peer review and any attached files.

Reviewer #1: No

Reviewer #2: Yes: Zhiying Deng

---

## [Author Response · Author response to Decision Letter 0]

21 Feb 2020

As requested, the phrase “data not shown” in the manuscript was deleted. Lines 250-252 were removed. Line 387 was replaced by reference to table S4. Lines 341-344 and lines 355 were replaced with respectively the exact p-values of the statistical test described in the Materials & Methods part (Lines 228-229) and the Pearson correlations coefficients.

Please find our full response to specific reviewer and editor comments in the letter "Response to Reviewers".

---

## [Decision Letter · Decision Letter 1]

6 Mar 2020

Wheat individual grain-size variance originates from crop development and from specific genetic determinism

PONE-D-19-34899R1

Dear Dr. Beral,

We are pleased to inform you that your manuscript has been judged scientifically suitable for publication and will be formally accepted for publication once it complies with all outstanding technical requirements.

With kind regards,

Aimin Zhang, Ph.D.

Academic Editor

PLOS ONE

Additional Editor Comments (optional):

Reviewers' comments:

Reviewer's Responses to Questions

**Comments to the Author**

1. If the authors have adequately addressed your comments raised in a previous round of review and you feel that this manuscript is now acceptable for publication, you may indicate that here to bypass the “Comments to the Author” section, enter your conflict of interest statement in the “Confidential to Editor” section, and submit your "Accept" recommendation.

Reviewer #1: All comments have been addressed

Reviewer #2: All comments have been addressed

2. Is the manuscript technically sound, and do the data support the conclusions?

Reviewer #1: (No Response)

Reviewer #2: Yes

3. Has the statistical analysis been performed appropriately and rigorously? 

Reviewer #1: (No Response)

Reviewer #2: Yes

4. Have the authors made all data underlying the findings in their manuscript fully available?

Reviewer #1: (No Response)

Reviewer #2: Yes

5. Is the manuscript presented in an intelligible fashion and written in standard English?

Reviewer #1: (No Response)

Reviewer #2: Yes

6. Review Comments to the Author

Reviewer #1: (No Response)

Reviewer #2: (No Response)

7. PLOS authors have the option to publish the peer review history of their article (what does this mean?). If published, this will include your full peer review and any attached files.

Reviewer #1: No

Reviewer #2: Yes: Zhiying Deng

---

## [Editor Report · Acceptance letter]

10 Mar 2020

PONE-D-19-34899R1 

Wheat individual grain-size variance originates from crop development and from specific genetic determinism 

Dear Dr. Beral:

I am pleased to inform you that your manuscript has been deemed suitable for publication in PLOS ONE. Congratulations! Your manuscript is now with our production department. 

With kind regards,

on behalf of

Prof. Aimin Zhang 

Academic Editor

PLOS ONE